# Optimization of Multilayered Walls for Building Envelopes Including PCM-Based Composites

**DOI:** 10.3390/ma13122787

**Published:** 2020-06-20

**Authors:** Victor D. Fachinotti, Facundo Bre, Christoph Mankel, Eduardus A. B. Koenders, Antonio Caggiano

**Affiliations:** 1Centro de Investigación de Métodos Computacionales (CIMEC), Universidad Nacional del Litoral (UNL), Consejo Nacional de Investigaciones Científicas y Técnicas (CONICET), Santa Fe 3000, Argentina; facubre@cimec.santafe-conicet.gov.ar; 2Institut für Werkstoffe im Bauwesen, Technische Universität Darmstadt, 64287 Darmstadt, Germany; mankel@wib.tu-darmstadt.de (C.M.); koenders@wib.tu-darmstadt.de (E.A.B.K.); caggiano@wib.tu-darmstadt.de (A.C.); 3Consejo Nacional de Investigaciones Científicas y Técnicas (CONICET), LMNI, INTECIN, Facultad de Ingeniería, Universidad de Buenos Aires, Ciudad Autónoma de Buenos Aires C1127AAR, Argentina

**Keywords:** thermal-energy storage (TES), phase change materials (PCMs), multilayered walls, building envelopes, non-linear optimization, genetic algorithms

## Abstract

This work proposes a numerical procedure to simulate and optimize the thermal response of a multilayered wallboard system for building envelopes, where each layer can be possibly made of Phase Change Materials (PCM)-based composites to take advantage of their Thermal-Energy Storage (TES) capacity. The simulation step consists in solving the transient heat conduction equation across the whole wallboard using the enthalpy-based finite element method. The weather is described in detail by the Typical Meteorological Year (TMY) of the building location. Taking the TMY as well as the wall azimuth as inputs, EnergyPlus^TM^ is used to define the convective boundary conditions at the external surface of the wall. For each layer, the material is chosen from a predefined vade mecum, including several PCM-based composites developed at the Institut für Werkstoffe im Bauwesen of TU Darmstadt together with standard insulating materials (i.e., EPS or Rockwool). Finally, the optimization step consists in using genetic algorithms to determine the stacking sequence of materials across the wallboard to minimize the undesired heat loads. The current simulation-based optimization procedure is applied to the design of envelopes for minimal undesired heat losses and gains in two locations with considerably different weather conditions, viz. Sauce Viejo in Argentina and Frankfurt in Germany. In general, for each location and all the considered orientations (north, east, south and west), optimal results consist of EPS walls containing a thin layer made of the PCM-based composite with highest TES capacity, placed near the middle of the wall and closer to the internal surface.

## 1. Introduction

According to the International Energy Agency (IEA), the buildings and construction sector across the world was responsible for 36% of final energy use and 39% of carbon dioxide (CO_2_) emissions in 2018 [1]. In Germany—and in Europe as well—buildings consume almost 40% of the total energy [2]. Far from Europe, in Argentina, a similar situation is faced: buildings are the largest energy consumers (33% of the total energy) [3].

Engineers, scientists, and decision makers all around the world are being appealed to promote best construction practices for using the least, cleanest, and/or most economical resources. The optimization of the energy efficiency in buildings plays a key role in this sense. It can be achieved by actuating on three main building components [4]: (i) building envelope (insulation, sealing, windows, and façades) [5,6]; (ii) technical building equipment (lighting, heating, and ventilation systems) [7,8]; and (iii) sustainable building systems (i.e., dealing with substantial aspects such as sustainable energies, automatization operations, and efficiency-centered planning) [9,10,11].

In this work, we are particularly interested in building envelopes, which are of key importance for controlling the heat loads in new buildings and/or retrofitted ones [12]. More particularly, we are interested in the use of phase change materials (PCMs) in building envelopes, making the so-called “energy dynamic building envelopes” [13]. These envelopes take advantage of the dynamic synergetic interplay between the thermal conductivity and the thermal-energy storage (TES) capacity of their components. 

PCMs can be directly added to construction materials of building envelopes as microencapsulated PCMs [14,15] or raw PCMs into carriers (e.g., impregnated gypsum boards [16] or porous aggregates [17]). Building envelopes can also contain macro-encapsulated PCMs having several shapes (plates, spheres, tubes, pipes, etc.) [18]. 

Usually, an energy dynamic building envelope is a multilayer wall. Most researchers place PCM-based layers either in contact [19,20] or very close to the building interior [21,22], without making any quantitative comparative analysis of position and thickness. However, Shi et al [23], who did compare the effect of placing the PCM-based layer innermost, outermost or in the middle of the wall, recommend to place it in the middle based on thermal comfort criteria for Hong Kong.

Thus, the placement and thickness of PCM-based layers remain an open question. Among other open questions, we aim to answer the four following questions: (i) When is it convenient to use PCMs? (ii) Which PCMs should be chosen? (iii) With which standard (non-PCM) materials should they be combined? (iv) How many layers are needed and/or how thick should the wall be in total? Any possible question has multiple answers. This work aims to offer a specific answer to each one while considering the performance index (energy efficiency, cost, environmental impact, etc.), the solar azimuth, and the local weather conditions along a typical year.

To determine the performance of a multilayer wall that is part of the building envelope, the heat exchange between the outer and inner environments through this wall must be known. To this end, we solve the transient heat conduction equation across the wall. This equation is generally highly nonlinear in the presence of PCMs due to the peaks in their effective heat capacity during latent heat release/absorption. Besides nonlinearity, another great challenge arises when a whole year must be considered. In this case, the external boundary condition varies on an hourly basis according to the typical meteorological year (TMY) of the building location. For the robust solution of such nonlinear equation at all the time steps, we use the enthalpy-based finite element method with implicit time integration [24].

Then, to optimize the performance of a multilayered wall in the building envelope, let each layer be made of a material consciously chosen from a predefined vade mecum of appropriate candidates, which especially include cementitious composites containing different fractions of PCMs either microencapsulated [25] or embedded into recycled brick aggregates (RBAs) [17], developed at the Institut für Werkstoffe im Bauwesen (WiB) of TU Darmstadt.

This vade mecum may also contain standard construction materials such as concrete and insulation materials (EPS, Rockwool, etc.), thus it can offer a wide choice of material properties for optimization purposes. Having a rich enough vade mecum, the evaluation of all the possible designs can be excessively expensive. Since we aim to continuously update and enlarge this vade mecum all along the recently launched four-year-long NRG-STORAGE project [26], we are obliged to use efficient optimization algorithms. Given its effectiveness when dealing with many integer design variables and discontinuities induced by such variables together with its low computational cost (mainly due to its easy parallelization), among other properties that are greatly appreciated in building performance simulation, and following our expertise in previous works [27,28,29], we decided to use genetic algorithms (GA).

The paper is structured as follows. Section 2 introduces the optimization-based methodology for the design of multilayered walls for building envelopes to be optimal in the light of a given performance index (as example, a purely thermal criterion is defined here). Section 3 describes the thermal modeling of a multilayered wall in a building envelope, pointing out the influence of local weather conditions on boundary conditions and determining the temperature as a function of the design. Section 4 is devoted to describing the vade mecum of materials for building envelopes, with emphasis on the PCM-based composites it contains. Section 5 shows and discuss the results of applying this methodology to walls with different orientations located in two different climates. Finally, Section 6 addresses the concluding remarks and future work.

## 2. Optimization-Based Design of Wallboards for Building Envelopes

Let us consider the *N*-layer wallboard with total thickness *L*. As shown in Figure 1, x∈[0,L] denotes the distance to the external surface and the layer i=1,2,…,L has thickness $\Delta x_{i}$ and lies between $x_{i-1}$ and $x_{i}=x_{i-1}+\Delta x_{i}$. 

To optimize the performance of the wall, each layer is allowed to be made of a different material. Let the integer $d_{i}$ identify the material at layer i=1,2,…,N. In the context of optimization, $d_{i}$ plays the role of a *categorical design variable* that can take a finite number of *levels*. Each level is an integer that identifies a candidate material. In general, each $d_{i}$ can take its own levels; in other words, the layer i can have its particular set of material choices; for instance, layer 1 (the external one) may only allow hydrophobic and non-flammable materials.

Each possible design of the N-layered wall is identified by the set $d=\left[d_{1},d_{2},\ldots ,d_{N}\right]$. Then, we propose to determine the optimal design, say $d^{\mathrm{opt}}$, by solving the following optimization problem (Equation (1)):(1)$$d^{\mathrm{opt}}=\arg\min C\left(d\right),$$
where C is the cost function representing the performance of the wall. 

Several choices for C can be found in the literature: the energy consumption for comfort in air-conditioned rooms and the degree-hours of discomfort in naturally-ventilated rooms, which can be considered either as multiple objectives [28,29] or combined in a weighted sum as a unique objective [27], the life cycle cost [30], the environmental impact [30,31], and a sum of the initial investment and the energy bill minus the resale value [32]. In these works, the cost function involves the whole building, but they can perfectly serve to characterize the performance of the wallboards enveloping these buildings.

In general, the problem in Equation (1) is subject to inequality constraints that serve to prescribe specific thresholds for the thermal transmittance (U-value) [32], the structural compliance, the weight of the whole wall, etc. The objective C and these may be interchangeable. For instance, one could either minimize the energy demand without exceeding a given budget or minimize the budget without exceeding a given energy demand.

For example, let us consider a purely thermal performance criterion adopting as cost function the total undesired heat loads all along the 8760 h of a typical meteorological year (TMY) (Equation (2)):(2)$$C\left(d\right)=\underset{\mathrm{undesired}\;\mathrm{gains}\;C_{\mathrm{gain}}}{\underbrace{\overset{8760}{\underset{h=1}{\sum }}h_{\mathrm{int}}\langle T_{\mathrm{surfint}}^{\left(h\right)}\left(d\right)-T_{\mathrm{tgt}}^{\max}\rangle }}+\underset{\mathrm{undesired}\;\mathrm{losses}\;C_{\mathrm{loss}}}{\underbrace{\overset{8760}{\underset{h=1}{\sum }}h_{\mathrm{int}}\langle T_{\mathrm{tgt}}^{\min}-T_{\mathrm{surfint}}^{\left(h\right)}\left(d\right)\rangle }}$$
where $h_{\mathrm{int}}$ is the heat convection coefficient between the wall and the indoor environment, $T_{\mathrm{surfint}}^{\left(h\right)}$ is the temperature at the internal wall surface at hour h, $T_{\mathrm{tgt}}^{\max}$ and $T_{\mathrm{tgt}}^{\max}$ are the maximum and minimum target (ideal) indoor temperatures (maybe time-dependent), and 〈u〉=(u+|u|)/2 is the ramp function such that there is no contribution to the undesired heat gains $C_{\mathrm{gain}}$ if $T_{\mathrm{int}}^{\left(h\right)}<T_{\mathrm{tgt}}^{\max}$ or to the undesired heat losses if $C_{\mathrm{loss}}$ if $T_{\mathrm{surfint}}^{\left(h\right)}>T_{\mathrm{tgt}}^{\min}$. In the above equation, $h_{\mathrm{int}}$, $T_{\mathrm{tgt}}^{\max}$, and $T_{\mathrm{tgt}}^{\min}$ are assumed to be given, while $T_{\mathrm{surfint}}^{\left(h\right)}$ is determined by the design d in a way to be defined in the next section.

## 3. Temperature Evolution Across the Wall

The temperature T at a distance x∈[0,L] from the external surface of the wall at the time instant t>0 is governed by the heat conduction Equation (3)
(3)$$\rho c_{\mathrm{eff}}\dot{T}-\frac{\partial }{\partial x}\;\left(k\frac{\partial T}{\partial x}\right)=0$$
subject to the initial conditions (Equation (4))
(4)$$T\left(x,0\right)=T_{0}$$
and the boundary conditions (Equations (5) and (6))
(5)$$k\frac{\partial T}{\partial x}=q_{\mathrm{ext}}\;\;\;\mathrm{at}\;x=0\;(\mathrm{external}\;\mathrm{surface})$$
(6)$$-k\frac{\partial T}{\partial x}=h_{\mathrm{int}}\left(T-T_{\mathrm{room}}\right)\;\;\;\;\mathrm{at}\;x=L\;\left(\mathrm{internal}\;\mathrm{surface}\right)$$

In the above equations $\dot{T}=\partial T/\partial t$ is the temperature rate, $\rho c_{\mathrm{eff}}$ is the effective heat capacity, k is the thermal conductivity, $q_{\mathrm{ext}}$ is the heat flux from the outdoor environment through the external wall surface (which evolves in time following the local weather conditions as described in Section 3.1), $h_{\mathrm{int}}$ is the heat convection coefficient at the internal wall surface (the same as in Equation (2)) and $T_{\mathrm{room}}$ is the indoor room temperature. 

Given the multilayered nature of the wall, the physical properties k and $\rho c_{\mathrm{eff}}$ are layer-wise defined: at a distance $x\in \left(x_{i-1},x_{i}\right)$ from the external surface, they are those of the material in layer i, that is the material $d_{i}$ in the vade mecum (see Figure 1).

In general, material properties depend on temperature (Equations (7)). Further, the effective heat capacity in PCMs is rate-dependent during phase changes [33]. Thus,
(7)$$\begin{array}{c} k=k\left(d_{i},T\right)\\ \rho c_{\mathrm{eff}}=\rho c_{\mathrm{eff}}\left(d_{i},T,\;\dot{T}\right) \end{array}\}\;\;\;\;\mathrm{at}\;x\in \left(x_{i-1},x_{i}\right)\;(\mathrm{layer}\;i)$$

The temperature dependence of material properties makes the heat Equation (3) nonlinear. This nonlinearity becomes severe in presence of PCMs due to the peaks in their heat capacity during phase changes. Then, it is crucial to develop a robust solver of the heat conduction Equation (3). Here, recourse is made to the enthalpy-based finite element formulation proposed by Morgan et al. [24]. For the sake of conciseness, the reader interested in the finite element implementation is referred to the just cited work.

Finally, since the thermal properties k and $\rho c_{\mathrm{eff}}$ at each layer i are dependent on $d_{i}$, it becomes apparent that the temperature T at any point x across the wall at any time instant t depends on the whole design $d=\left[d_{1},d_{2},\;\ldots ,d_{N}\right]$, namely T=T(x,t,d). Particularly, the temperature at the internal surface at the hour h is $T_{\mathrm{surfint}}^{\left(h\right)}\left(d\right)=T\left(L,h\;\mathrm{hours},d\right)$, making explicit the influence of the design d on the performance of the wall as represented by the cost function C(d) given by Equation (2). 

### 3.1. External Boundary Conditions

The heat flux from the outdoor environment is defined as follows (Equations (8)):(8)$$q_{\mathrm{ext}}\left(t\right)=\alpha q_{\mathrm{solar}}\left(t\right)+h_{\mathrm{ext}}\left(t\right)\left(T_{\mathrm{out}}\left(t\right)-T_{\mathrm{surfext}}\left(t\right)\right)$$
where $q_{\mathrm{solar}}$ is the absorbed short-wave (direct and diffuse) solar radiation, α is the solar absorbance (assumed equal to $0.6\;h_{\mathrm{ext}}$ is the external heat transfer coefficient $T_{\mathrm{out}}$ is the outdoor temperature, and $T_{\mathrm{surfext}}\left(t\right)=T\left(0,t,d\right)$ is the temperature at the external surface.

The heat flux $q_{\mathrm{ext}}$ depends on t not only via $T_{\mathrm{surfext}}\left(t\right)$ but also through $T_{\mathrm{out}}$, $h_{\mathrm{ext}}$ and $q_{\mathrm{solar}}$, which change following the instantaneous local weather conditions as defined by the Typical Meteorological Year (TMY). 

Regarding $T_{\mathrm{out}}$, this coincides with either the dry bulb temperature $T_{\mathrm{db}}$ for $h_{\mathrm{ext}}<1000\;W/(m^{2}K)$ or the wet-bulb temperature $T_{\mathrm{wb}}$ for $h_{\mathrm{ext}}=1000\;W/(m^{2}K)$, with $T_{\mathrm{db}}$ and $T_{\mathrm{wb}}$ directly taken from the TMY, where they vary on an hourly basis. On the other hand, $q_{\mathrm{solar}}$ and the convection coefficient $h_{\mathrm{ext}}$ are computed using EnergyPlus™ [34]. To take into account not only the local weather conditions but also the influence of the surface facing angle on these variables, EnergyPlus is applied to a square 4-m-wide, 4-m-deep, and 3-m-high thermal zone with walls facing north, east, south, and west. Regarding $h_{\mathrm{ext}}$, it is computed using the so-called *AdaptiveConvectionAlgorithm* in EnergyPlus, which takes into account the wind direction and magnitude and sets $h_{\mathrm{ext}}$ to an arbitrarily high value ($1000\;W/(m^{2}K))$ at the wall exposed to the wind when it is raining, forcing to assume $T_{\mathrm{out}}=T_{\mathrm{db}}$ at that instant.

## 4. Vade Mecum of Materials for Building Envelopes

To improve the performance of a multilayer wall, let each layer be built of a material thoughtfully chosen from the henceforth called vade mecum of materials for building envelopes. This is a database that should contain a wide choice of materials in terms of thermal properties (conductivity, specific heat, and thermal energy storage capacity, etc.) as well as non-flammability, water and air tightness, weight, cost, and embedded energy, among others, such that it offers a large enough design space for optimization purposes. Mathematically speaking, each material in the vade mecum is a level of the categorical design variable $d_{i}$. Note that a vade mecum containing M materials gives rise to $N^{M}$ different designs for an N-layered wall. 

As a first step, the current vade mecum is built on the basis of purely thermal criteria. Particularly, we are interested in using two cement-based PCM-composites, henceforth referred to as MPCM-p and RBA-p, with p related to the PCM content. 

The MPCM-p is a concrete with w/c=0.45, 70 vol% of normal aggregates (granitic crushed stones) and 30 vol% of a PCM paste containing p=0, 10, or 20 vol% of microencapsulated PCM. The PCM is Micronal^®^ DS 5038 X type, which is a powder of microencapsulated paraffin wax developed by BASF, with a melting point of around 26 °C and a heat storage capacity of 145 kJ/kg [35]. The effective heat capacity of MPCM-p mixtures as distinct temperature-dependent functions for either heating or cooling are shown on the left of Figure 2 (see [25] for more details on these mixtures). 

The RBA-p is a concrete with w/c=0.50, 30 vol% of surrounding cement paste, and 70 vol% of recycled brick aggregates (RBA), which are filled with p= 0, 65, and 80 vol% of PCM. In this case, the PCM is the non-encapsulated paraffin wax RT 25 HC^®^ developed by RUBITHERM^®^, with a melting point of around 25 °C and a heat storage capacity of 210 kJ/kg [36]. The effective heat capacity of MPCM-p mixtures are also temperature-dependent functions that differ for heating and cooling, and are depicted on the right of Figure 2. More details on RBA-p composites are given in [17].

To enlarge the choice of thermal properties, the vade mecum also contains two widely used insulating materials: rockwool and expanded polystyrene (EPS). The current vade mecum is summarized in Table 1. Let us note that this is a first version, to be continuously enriched with additional properties (cost, embedded energy, etc.) and enlarged all along the recently launched project NRG-STORAGE, mainly to account for other performance indexes (cost, environmental aspects, etc.). Further, the wall must exhibit fire safety and water tightness, among other essential requirements for real life applications. Considering fire safety for instance, PCM-based concretes offer a non-flammable choice; on the contrary, EPS and wools exhibit a high flammability, which is a well-known critical issue yet to be solved. Thus, although indispensable, these additional requirements may be detrimental to the performance of the wall; therefore, in the optimization problem in Equation (1), they are not represented by the cost function C but by inequality constraints. 

## 5. Numerical Results

This section reports the results of the optimization of multilayered systems for external walls, considering four different orientations (N, S, W, and E) and two different locations (Sauce Viejo in Argentina and Frankfurt in Germany). Sauce Viejo has a humid subtropical climate, which is considered as Cfa in the Köppen–Geiger classification; the average temperatures along typical summer and winter weeks are 24.4 and 13.8 °C, respectively. Frankfurt has a warm temperate climate, Cfb in the Köppen–Geiger classification, with average temperatures of 18.0 and 1.7 °C, respectively, for typical summer and winter weeks. For later discussions concerning orientation, let us keep in mind that Sauce Viejo and Frankfurt are in the Southern and Northern hemispheres, respectively.

For an accurate evaluation of the performance of a building envelope all along a year, the weather at each location is described by its typical meteorological year (TMY). This TMY is a database with relevant weather variables (including dry-bulb and dew-point temperatures, wind speed and direction, relative humidity, total sky cover, ceiling height, atmospheric pressure, global horizontal solar radiation, diffuse and direct normal solar radiation, precipitation, etc.) given on an hourly basis along twelve typical meteorological months, which were chosen from different years following the Sandia method [37]. The TMYs for Sauce Viejo and Frankfurt can be downloaded for free from Climate.OneBuilding [38]; that at Sauce Viejo was recently generated by Bre and Fachinotti [39].

As pointed out in Section 2, the cost function C(d) (to be minimized) represents the undesired heat loads along a TMY defined by Equation (2). We further assume that: (i) the room temperature $T_{\mathrm{room}}$ appearing in the boundary condition at the internal surface (Equation (6)) is ideally maintained at 24 °C; (ii) the maximal and minimal temperatures involved in the definition of C(d) (Equation (2)) are set to the same value, i.e., $T_{\mathrm{tgt}}^{\min}\equiv T_{\mathrm{tgt}}^{\min}\equiv T_{\mathrm{room}}=24\;{}^{\circ}C$; and (iii) the convection coefficient of the internal surface in Equations (2) and (6) is set to the typical value $h_{\mathrm{int}}=8.24\;W/\left(m^{2}K\right)$, as adopted by Biswas et al. [21,22] for non-reflective vertical interior walls.

### 5.1. Reference Solutions on Homogenous Walls

For comparison purposes, let us start by considering a homogeneous 20-cm-thick external wall made of one of the candidate materials found in the vade mecum (Table 2). 

The transient heat conduction Equation (3) was solved using the enthalpy-based finite element method [24] with Euler-backward (implicit) time stepping. A previous analysis was carried out to determine the best deal between and computational cost, from which we decided to use four linear finite elements per layer and constant time steps of half-an-hour (i.e., 17,520 time steps along the TMY). To take due account of the steep variation of the effective heat capacity in PCM-based composites, six Gauss points were used in the corresponding finite elements, while two Gauss points (as usual) were used in the remaining finite elements. 

Then, 64 problems were solved, one per each location, material, and orientation. From now on, for the sake of simplicity, let us refer simply as N to the case of a wall facing N, and so on for E-, S, and W-facing cases.

At Sauce Viejo, as shown in Figure 3, the undesired heat gains $C_{\mathrm{gain}}$ and losses $C_{\mathrm{loss}}$ are balanced. Using the PCM–concrete composites, $C_{\mathrm{gain}}$ is maximal for the N and minimal for S; for W and E, it is almost as prejudicial as for N. Regarding $C_{\mathrm{loss}}$, its maximum and minimum occur at S and N, respectively. Once again, E and W are closer to the worst case (S). In general, considering the total undesired loads $C=C_{\mathrm{gain}}+C_{\mathrm{loss}}$, the worst case is W, followed by E, N and S, in that order. Using insulating materials (either rockwool or EPS), the total C considerably reduced. Best performances are achieved using EPS (the least conductive material of the vade mecum). For EPS, C attains its minimum value for S.

At Frankfurt, as shown in Figure 4, $C_{\mathrm{loss}}$ is considerably higher than $C_{\mathrm{gain}}$. This made the conclusions for $C_{\mathrm{loss}}$ also valid for the total heat loads C. By using the PCM–concrete composites, the performance of the wall is not as sensitive to the orientation as it is for Sauce Viejo. In general, N is the worst case, but E, W and S are not much better. Once again, the performance is greatly improved by using insulating materials.

### 5.2. Optimal Multilayered Walls

Given the 20-cm-thick external wall of the preceding section, let it be made of 20 equally thick layers, and each layer is allowed to be made of one the eight materials in the vade mecum (Table 1). In this case, a design is actually a stacking sequence defined by $d=\left[d_{1},d_{2},\;\ldots ,d_{20}\right]$, where $d_{i}$ identifies the material in layer i. 

Then, the optimal stacking sequence is determined by solving the nonlinear, integer programming problem given by Equation (1) by using genetic algorithms (GA). Note that a homogeneous wall made of the material m in the vade mecum represents a possible design d of the current multilayered wall, where all the layers are made of the same material m, i.e., $d_{i}=m=constant$ for i=1,2,…,N. The best homogeneous wall from the previous section (that made of EPS for all the cases) has been included in the initial population for GA.

The optimal solutions for both locations and the four orientations are shown in Table 2. Not surprisingly, only two materials, among the eight possible ones in the vade mecum, are present in all the optimal solutions: that with the lowest conductivity (level 8 = EPS) and that with the highest effective thermal energy storage capacity (level 6 = RBA-80). The great majority of the layers are made of EPS (16 or 18 for Sauce Viejo and 19 for Frankfurt). 

At Sauce Viejo (see Figure 5), despite the little use of RBA-80, the energy performance of the optimal multilayered wall is considerable better than that of the EPS (insulating only) wall: C is reduced between 23.4% and 45.6%, bounds corresponding to S and N, respectively. 

At Frankfurt, as shown in Figure 6, only Layer 11 is made of RBA-80 for all the considered orientations. For this optimal wall, $C_{\mathrm{gain}}$ is greatly reduced (from 70.6% for W to 78.9% for N). However, the weight of $C_{\mathrm{gain}}$ into C is considerably lower than that of $C_{\mathrm{loss}}$, for which there is a modest improvement: from 0.4% for N to 8.3% for S. At the end, C is reduced between 4.8% for N to 18.8% for S.

In general, for Sauce Viejo (in the Southern hemisphere) as well as for Frankfurt (in the Northern one), the most beneficial effect of RBA-80 is observed at the most sun-exposed wall. Furthermore, the use of RBA-80 leads to considerably higher improvements in Sauce Viejo, where the temperatures remain a longer time in the phase-change range of RBA-80 (from 20 °C to 26 °C, approximately).

Finally, let us remark that Table 2 is the initial version of a new vade mecum of multilayered wall systems for building envelopes. For the time being, it offers a quick answer to the question of minimizing the undesired thermal loads at locations climatically close to those analyzed here. Once again, it is a goal of the recently launched NRG-STORAGE project to continuously enlarge this vade mecum to offer quick solutions for more climates considering different performance indexes.

## 6. Conclusions

This article introduces an optimization-based methodology to improve the performance of multilayer walls to be used in building envelopes. This is done in the following steps:A vade mecum of materials is built to be used in building envelopes, including particularly materials with thermal energy storage capacity and insulating properties, which should offer a wide choice of material responses.Given the location and the wall azimuth, EnergyPlus is used to translate the local typical meteorological year (TMY) into time-dependent boundary conditions for the heat conduction equation.Given a multilayered wall “design”, i.e., a specific stacking sequence of materials chosen from the vade mecum of Step i, the heat conduction equation, subject to the boundary conditions from Step ii, is solved along a whole Typical Meteorological Year (TMY) using a robust finite element method to determine the temperature across a given multilayered wall.The temperature evolution at the internal wall surface, resulting from Step iii, serves to determine the thermal performance index of the current design, here defined by a cost function representing the undesired thermal loads along a TMY.Genetic algorithms are used to make the designs evolve until achieving optimal performance.Steps ii–v must be repeated first for the remaining orientations at the same location and then for different locations.

Here, the methodology was applied to optimize thermal performance of envelopes in Sauce Viejo (Argentina) and Frankfurt (Germany), having humid subtropical and warm temperate climates, respectively. Further, the walls were assumed to face north, east, south, and west. Considering a 20-cm-thick external wall, the optimal solution in any case is mostly made of EPS (the best insulating material in the current vade mecum), including a 1–4-cm-thick layer of a PCM-based composite (the material with the highest thermal energy storage capacity in the current vade mecum). In general, This PCM-based layer is placed next to the middle of the wall, closer to the internal surface. In this way, this methodology defines not only the proper placement but also the proper thickness of the PCM-based layer considering weather and orientation.

Furthermore, despite the little use of PCMs, the undesired heat loads were reduced in comparison to a 20-cm-thick EPS wall: up to 18.8% and 45.6% for Frankfurt and Sauce Viejo, respectively. The better performance at Sauce Viejo is explained by the fact that local temperatures remain for longer periods closer to the phase-change temperatures of the PCM-based composites available in the current vade mecum. 

Further steps which follow from this research will address the enrichment of the vade mecum, including particularly PCM-based composites with various phase-change temperature ranges and materials. 

Once the vade mecum of materials proves to be wide enough, optimal solutions can be obtained by applying the current methodology within a wide range of climates and different performance indexes, which will be taken as inputs for a new vade mecum of building envelopes.

## Figures and Tables

**Figure 1 materials-13-02787-f001:**
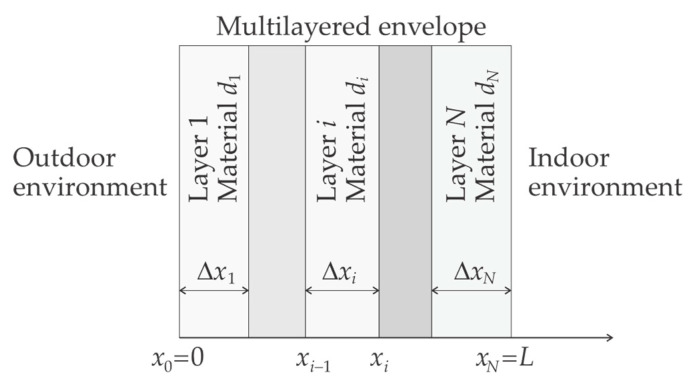
Schema of a building envelope using a multilayered wall.

**Figure 2 materials-13-02787-f002:**
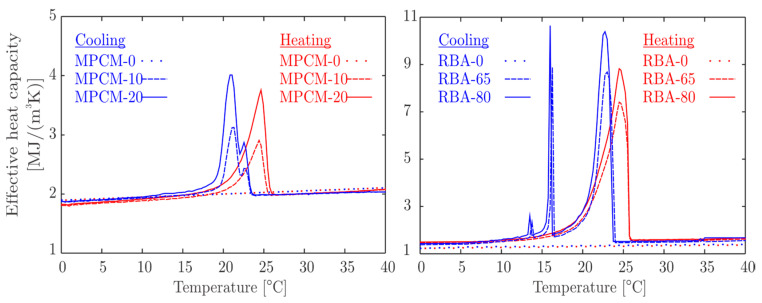
Effective heat capacities of MPCM-p concretes (on the **left**) and RBA-p concretes (on the **right**).

**Figure 3 materials-13-02787-f003:**
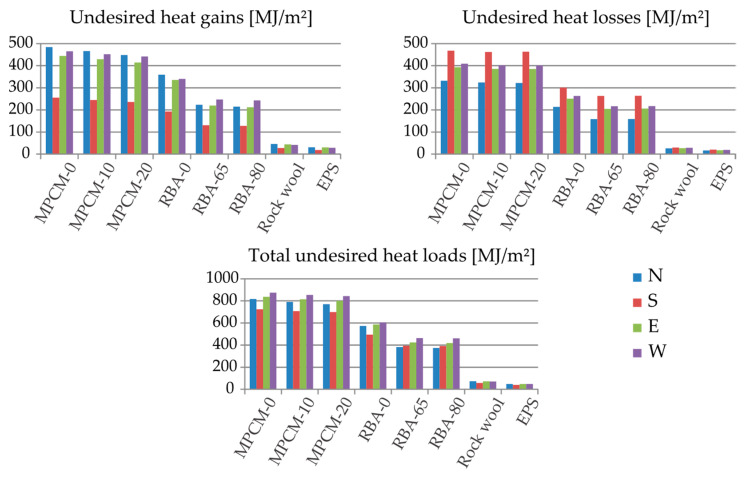
Undesired heat gains, heat losses and total heat loads in 20-cm-thickness external walls made of different materials at Sauce Viejo (Argentina) for different orientations.

**Figure 4 materials-13-02787-f004:**
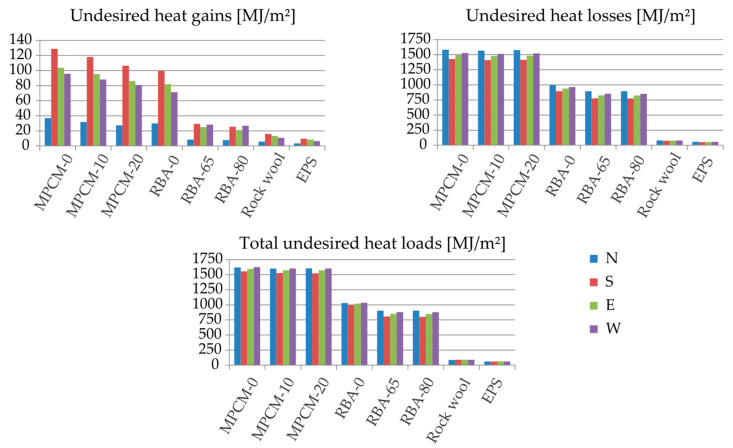
Undesired heat gains, heat losses and total heat loads in 20-cm-thickness external walls made of different materials at Frankfurt (Germany) for different orientations.

**Figure 5 materials-13-02787-f005:**
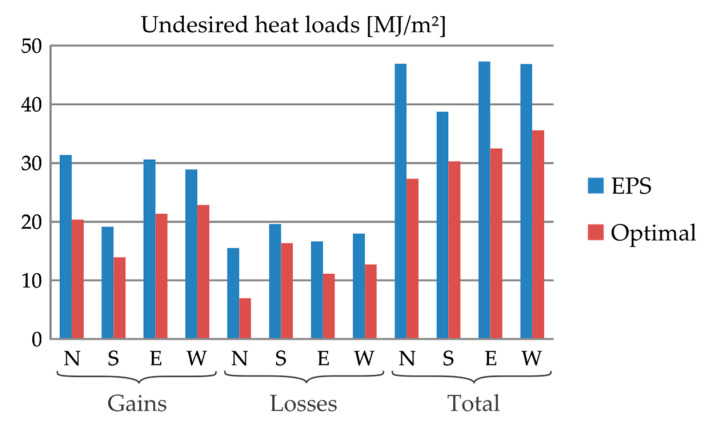
Undesired heat loads in a 20-cm-thick external wall at Sauce Viejo (Argentina) for different orientations: comparison between the EPS insulating wall and the optimal multilayered wall.

**Figure 6 materials-13-02787-f006:**
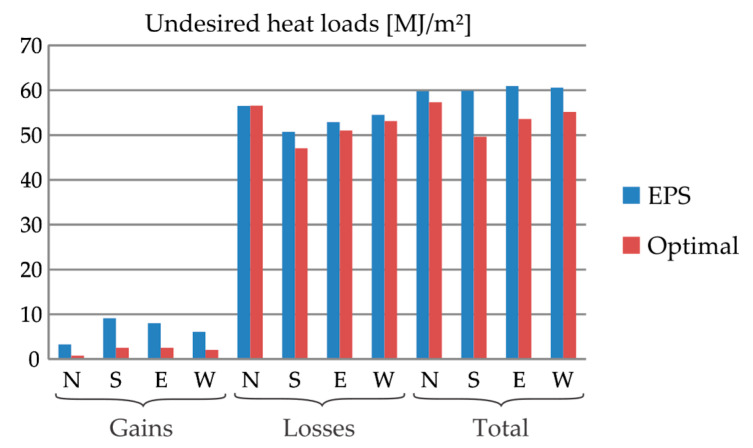
Undesired heat loads in a 20-cm-thick external wall at Frankfurt (Germany) for different orientations: comparison between the EPS insulating wall and the optimal multilayered wall.

**Table 1 materials-13-02787-t001:** Vade mecum of materials for building envelopes.

Level	Name	Effective Thermal Conductivity [W/(mK)]	Effective Heat Capacity [J/(m^3^K)]
1	MPCM-0	2.336	Temperature dependent *
2	MPCM-10	2.311	Temperature dependent *
3	MPCM-20	2.311	Temperature dependent *
4	RBA-0	0.910	Temperature dependent *
5	RBA-65	0.769	Temperature dependent *
6	RBA-80	0.769	Temperature dependent *
7	Rockwool	0.042	33,750
8	EPS	0.030	60,000

* see Figure 2.

**Table 2 materials-13-02787-t002:** Vade mecum of multilayered systems for building envelopes.

Location	Orientation	Layer *																			
		1	2	3	4	5	6	7	8	9	10	11	12	13	14	15	16	17	18	19	20
Sauce Viejo	N	8	8	8	8	8	8	8	8	8	6	6	6	6	8	8	8	8	8	8	8
	S	8	8	8	8	8	8	8	8	8	6	6	8	8	8	8	8	8	8	8	8
	E	8	8	8	8	8	8	8	8	8	8	8	8	6	6	8	8	8	8	8	8
	W	8	8	8	8	8	8	8	8	8	8	6	6	8	8	8	8	8	8	8	8
Frankfurt	N	8	8	8	8	8	8	8	8	8	8	6	8	8	8	8	8	8	8	8	8
	S	8	8	8	8	8	8	8	8	8	8	6	8	8	8	8	8	8	8	8	8
	E	8	8	8	8	8	8	8	8	8	8	6	8	8	8	8	8	8	8	8	8
	W	8	8	8	8	8	8	8	8	8	8	6	8	8	8	8	8	8	8	8	8

* Layers 1 and 20 are the outermost and the innermost, respectively. Materials 6 and 8 are RBA-80 and EPS, respectively.

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
