# Peer review of "Optimization of Multilayered Walls for Building Envelopes Including PCM-Based Composites"

_materials, 2020, doi:10.3390/ma13122787_

Round 1

Reviewer 1 Report

Dear Authors,

The topic "Optimization of multilayered walls for building 2 envelopes including PCM-based composites" seems to be interesting specially in countries in which energy consumption has a significant role. The manuscript prepared in a good way however it could be improved by the following comments:

1) The introduction is too long. it would be better to cut from line 95 and add the text to a new section which could call Analysis procedure or something else.

2) The wall simulated with only 2 different materials however the variety of such construction would be totally different.

3) The quality of all graphs (figures) should improve.

4) The whole results should verify by physical model or real construction. The lack of real data is clear on the paper.

5) Some of the names written in another font which I'm not sure if it fits the Journal format (like line 23-24, 111, ...)

6) There are lots of figures, equations, and tables citation which missed during the preparation of the manuscript (like lines 105-106, 142-143, 146, 160, 186, ...)

Reviewer 2 Report

Please check the cited references. Twenty "Error! Reference source not found" are present in the paper text.

Author Response

Point 1: Please check the cited references. Twenty "Error! Reference source not found" are present in the paper text.

Response 1: We would like to thank the reviewer for making us realize of these errors. They did not exist in the .doc file we uploaded, being apparently produced when the journal submitting system converted it into .pdf. We will warn the editor to fix this bug.

Reviewer 3 Report

This paper presents results from an experimental program aimed to develop a method for optimizing multilayered walls via the use of PCM composites. The presented idea is interesting. The following items are to be addressed before the manuscript can be published:

  • Typos exist across the manuscript. Especially those related to incorrect citations i.e. “Error! Reference source not found”
  • How economical is the proposed system? As compared to commercially available products.
  • Heat transfer is primarily a function of thermal properties. Please elaborate a bit on the density of the system. How sensitive is this property to temperatures?
  • Can the proposed insulation system also work as a fire insulation? The works of Hawileh et al. show the use of gypsum-based system can be effective. Perhaps this point can be highlighted.

    This paper presents results from an experimental program aimed to develop a method for optimizing multilayered walls via the use of PCM composites. The presented idea is interesting. The following items are to be addressed before the manuscript can be published:

    • Typos exist across the manuscript. Especially those related to incorrect citations i.e. “Error! Reference source not found”
    • How economical is the proposed system? As compared to commercially available products.
    • Heat transfer is primarily a function of thermal properties. Please elaborate a bit on the density of the system. How sensitive is this property to temperatures?
    • Can the proposed insulation system also work as a fire insulation? The works of Hawileh et al. show the use of gypsum-based system can be effective. Perhaps this point can be highlighted.

Round 2

Reviewer 3 Report

Thank you!